# Ligand coupling mechanism of the human serotonin transporter differentiates substrates from inhibitors

Ralph Gradisch ⬤[1], Katharina Schlögl[2], Erika Lazzarin ⬤[1], Marco Niello ⬤[1,3], Julian Maier ⬤[1], Felix P. Mayer ⬤[4,5,8], Leticia Alves da Silva[1], Sophie M. C. Skopec[1], Randy D. Blakely ⬤[4,5], Harald H. Sitte ⬤[1,6,7], Marko D. Mihovilovic ⬤[2] & Thomas Stockner ⬤[1] ✉

The presynaptic serotonin transporter (SERT) clears extracellular serotonin following vesicular release to ensure temporal and spatial regulation of serotonergic signalling and neurotransmitter homeostasis. Prescription drugs used to treat neurobehavioral disorders, including depression, anxiety, and obsessive-compulsive disorder, trap SERT by blocking the transport cycle. In contrast, illicit drugs of abuse like amphetamines reverse SERT directionality, causing serotonin efflux. Both processes result in increased extracellular serotonin levels. By combining molecular dynamics simulations with biochemical experiments and using a homologous series of serotonin analogues, we uncovered the coupling mechanism between the substrate and the transporter, which triggers the uptake of serotonin. Free energy analysis showed that only scaffold-bound substrates could initiate SERT occlusion through attractive long-range electrostatic interactions acting on the bundle domain. The associated spatial requirements define substrate and inhibitor properties, enabling additional possibilities for rational drug design approaches.

Serotonin (5-hydroxytryptamine, 5HT) is a biogenic amine, which plays important modulatory roles that impact neuronal activity and neuropsychological processes[1]. Imbalances within monoaminergic signalling are associated with a variety of neuropsychiatric diseases, comprising e.g. major depressive disorder (MDD), migraine, anxiety-related disorders, autism or attention-deficit/hyperactivity disorder[2–6]. According to the World Health Organization, MDD is the leading disorder affecting more than 250 million people worldwide[7]. Although the pathophysiological mechanisms are not fully understood, involvement of the serotonergic system is a suggested shared feature[8–10]. 5HT signalling is temporally shaped, spatially confined and terminated by the 5HT transporter (SERT, SLC6A4) through active re-uptake into the presynaptic neuron[11–13]. Drugs interacting with SERT interfere with its physiological function and increase extracellular 5HT levels[4,14]. Typical inhibitors include frequently prescribed drugs, such as S-citalopram (S-Cit) and paroxetine, but also include abused drugs such as cocaine (COC). These rather large molecules act as competitive inhibitors that wedge between the membrane-anchored scaffold domain and the mobile bundle domain of outward-facing SERT[15,16], thereby preventing occlusion of SERT as an early step in the transport cycle[17]. Atypical inhibitors such as ibogaine (IBO) act as non-competitive inhibitors by stabilising the inward-facing

[1]Medical University of Vienna, Institute of Physiology and Pharmacology, Waehringer Straße 13A, 1090 Vienna, Austria. [2]TU Wien, Institute of Applied Synthetic Chemistry, Getreidemarkt 9, 1060 Vienna, Austria. [3]Genetics of Cognition Laboratory, Neuroscience area, Istituto Italiano di Tecnologia, via Morego, 30, 16163 Genova, Italy. [4]Florida Atlantic University, Department of Biomedical Science, Jupiter, FL 33458, USA. [5]Stiles-Nicholson Brain Institute, Jupiter, FL 33458, USA. [6]Al-Ahliyya Amman University, Hourani Center for Applied Scientific Research, Amman, Jordan. [7]Medical University of Vienna, Center for Addiction Research and Science, Waehringer Straße 13A, 1090 Vienna, Austria. [8]Present address: Department of Neuroscience, Faculty of Health and Medical Sciences, University of Copenhagen, DK-2200 Copenhagen, Denmark. ✉e-mail: thomas.stockner@meduniwien.ac.at

conformation[18–20]. Both types of inhibitors prevent the progression through the transport cycle. Mechanistically different, amphetamines and their derivatives are substrates of SERT that induce efflux of the cognate intracellular substrate by reverse transport[21]. Among them, the illicit drug 3,4-methylenedioxy-N-methylamphetamine (MDMA)[22,23] has recently shown therapeutic potential and entered Phase III clinical trials for the treatment of post-traumatic stress disorder[22,24,25].

SERT is a secondary active transporter that utilises the transmembrane ion gradient for energising substrate re-uptake[26,27]. 5HT binds to the central substrate binding site (S1) of ion-bound SERT in the outward-open conformation. Conformational rearrangements, specifically a tilting motion of the flexible bundle domain comprising TM1, 2, 6 and 7 towards and away from the membrane-anchored rigid scaffold domain (TM3, 4, 8 and 9), allow the transition to the inward-facing conformation associated with cargo release to the cytosol. Kinetic schemes have defined substrate and ion stoichiometry and modelled the steps of the transport cycle[26,28–32]. Static structural data have resolved the most stable conformations in the presence of an antibody and ligand, which are the outward-open (OO)[29], partially-occluded (PO)[18], fully-occluded (OC)[28], and inward-open (IO)[28] states, but knowledge of conformational dynamics and the free energy surface remained limited[33–35], and therefore key aspects of the coupling between substrate binding and transport remains to be defined. We have previously shown that binding of Na$^+$ to SERT is fast (~9.8·10$^6$ M$^{-1}$·s$^{-1}$)[32,36], that bound Na$^+$ stabilises the outward-open conformation[37] and provided evidence that interactions of 5HT with the bundle domain may play an important role for initiating transport cycle[34].

In the present study, we systematically investigate the mechanism of coupling between substrates and human SERT that initiates the transport cycle as it transits to an occluded state. Initially, we combine unbiased molecular dynamics simulations (120 μs total simulation time) with analyses of the conformational dynamics and their associated energies to obtain an in-depth in silico description of the effects mediated by a series of 5HT homologues. In the second step we synthesise these compounds and characterise them in vitro as well as ex vivo. We reveal that these molecules display strikingly distinct effects by differentially interfering with the coupling between the compound and the transporter. Our results elucidate key molecular details responsible for initiating substrate-triggered transport and lay a foundation for the future rational design of innovative SERT inhibitors by unravelling the chemical and geometrical requirements of substrate-transporter coupling.

## Results

### Interactions of substrate with the bundle domain are critical for SERT occlusion

Previous computational data suggested that the cognate substrate 5HT might induce SERT occlusion by fitting into the substrate binding site S1 in the occluded state of SERT[34], while being slightly too small for the S1 in the outward-open state. This prompted us to speculate that interactions of 5HT's alkylamine moiety with the bundle domain were important for SERT occlusion, as 5HT initially interacts with D98 (TM1) and subsequently with the backbone oxygen of F335 (TM6). This interaction pattern is observed in structural as well as computational studies (Fig. 1a, c)[18,29,34].

We inferred from these data that the position of the positively charged nitrogen of 5HT would be critical for triggering occlusion and hypothesised that changing the position of the nitrogen would interfere with the mechanism of coupling between the substrate and SERT. This hypothesis was supported by the D98E substitution in SERT that resulted in a decreased 5HT transport capacity, but was accompanied with an uncoupling of extracellular Na$^+$ and Cl$^-$[38]. Accordingly, instead of mutating the target protein, we modified the length of the alkyl-linker to change the position of the nitrogen (Fig. 1d) and created

methylamine-5HT (M-5HT), which shortens the chain by one methylene group thus moving the amine ~ 0.13 nm closer to the indole. Extending the alkyl chain by one or two methylene groups results in propylamine-5HT (P-5HT) and butylamine-5HT (B-5HT), respectively, which moves the nitrogen by ~ 0.12 nm and ~ 0.25 nm away from the rigid ring system. Ten independent systems for each of the apo- and the compound-bound outward-open SERT (5HT and its analogues M-5HT, P-5HT, B-5HT and the well characterised SERT blocker cocaine) were inserted in a membrane mimicking the physiological environment, comprising 1-palmitoyl-2-oleoyl-glycero-3-phosphocholine (POPC) and cholesterol in a 70:30 ratio. All these systems were independently equilibrated and simulated for 2 μs, cumulating to a total simulation time of 120 μs of unbiased molecular dynamics (MD) simulation.

The degree of SERT occlusion was assessed by measuring the distances between TM1b (residue L99 to Q111) to TM9up (residue F475 to S477) (SI Fig. 1) and TM6a (residue G324 to L337) to TM9up across the outer vestibule (Fig. 1e). All ten trajectories of M-5HT remained in an outward-open state. The small decrease in distances is consistent with sensing the presence of a ligand by SERT. Of the 5HT-bound SERT systems, 3 out of 10 trajectories reach the occluded state, while for only 1 out of the 10 simulations of both P-5HT-bound as well as B-5HT-bound SERT systems the occluded state is reached. As expected, the cocaine bound systems could not reach the occluded state, because they were sterically blocked by the larger inhibitor. SERT remained in a wide outward-open state in the absence of ligands[34,37].

To collectively analyse the conformational dynamics of the entire dataset of all 50 ligand-bound and 10 ligand-free simulations, we conducted principal component analysis (PCA) on the Cα-atoms of the 12 TM helices. The first eigenvector or principal component 1 (PC1) represents the largest motion and describes the structural rearrangements of the bundle domain that tilts towards the scaffold domain, while the second principal component (PC2) shows a rotation of the bundle domain perpendicular to PC1 (Fig. 2a). Projecting the trajectories for each ligand (Fig. 2b) on the same PC1 and PC2 coordinates allowed us to globally compare the substrate mediated motions. Projections of the M-5HT simulations showed that SERT remains outward-open, as motions of occlusion (described by PC1) are limited. In contrast, projection of the 5HT-bound trajectories showed extensive motions along the axis of SERT occlusion. Also, P-5HT-bound SERT shows a comparable extent of motion along PC1, but the density indicates that the occluded state is less populated, in accordance with distances across the vestibule (Fig. 2b). Projections of B-5HT-bound SERT trajectories suggested that the occluded state is more difficult to reach, indicated by an enhanced population of an outward-open like conformation. As expected, the trajectories of the cocaine-bound SERT are stabilised in an outward-open conformation. The ligand-free SERT remained wide outward-open and showed larger motions, which is consistent with increased dynamics in absence of ligands.

To assess the impact of interactions of a compound with the bundle domain and the ability of a compound to trigger transport by inducing SERT occlusion, we correlated the interaction energy (VdW + electrostatic) between compounds and residue F335 on the bundle domain with the openness of outer vestibule as quantified by the distance of TM6a (bundle domain) to TM9up (scaffold domain) (Fig. 2c, d). The corresponding plots for residue D98 (TM1b) are shown in Supplementary Fig. 1. These 2D plots suggest that M-5HT is unable to induce SERT occlusion in the time window of 2 μs, despite repeated strong interactions with F335. Most likely, M-5HT is too short to maintain a continuous pulling force, which is needed for SERT occlusion. As a consequence, this allows the bundle domain to relax back to the outward-open conformation. In contrast, 5HT induced SERT occlusion once it engaged with strong interaction with F335. P-5HT has a smaller ability to trigger SERT occlusion despite showing strong interactions with D98 and F335. Due to its increased size, it can already form strong interactions with limited movements of the bundle

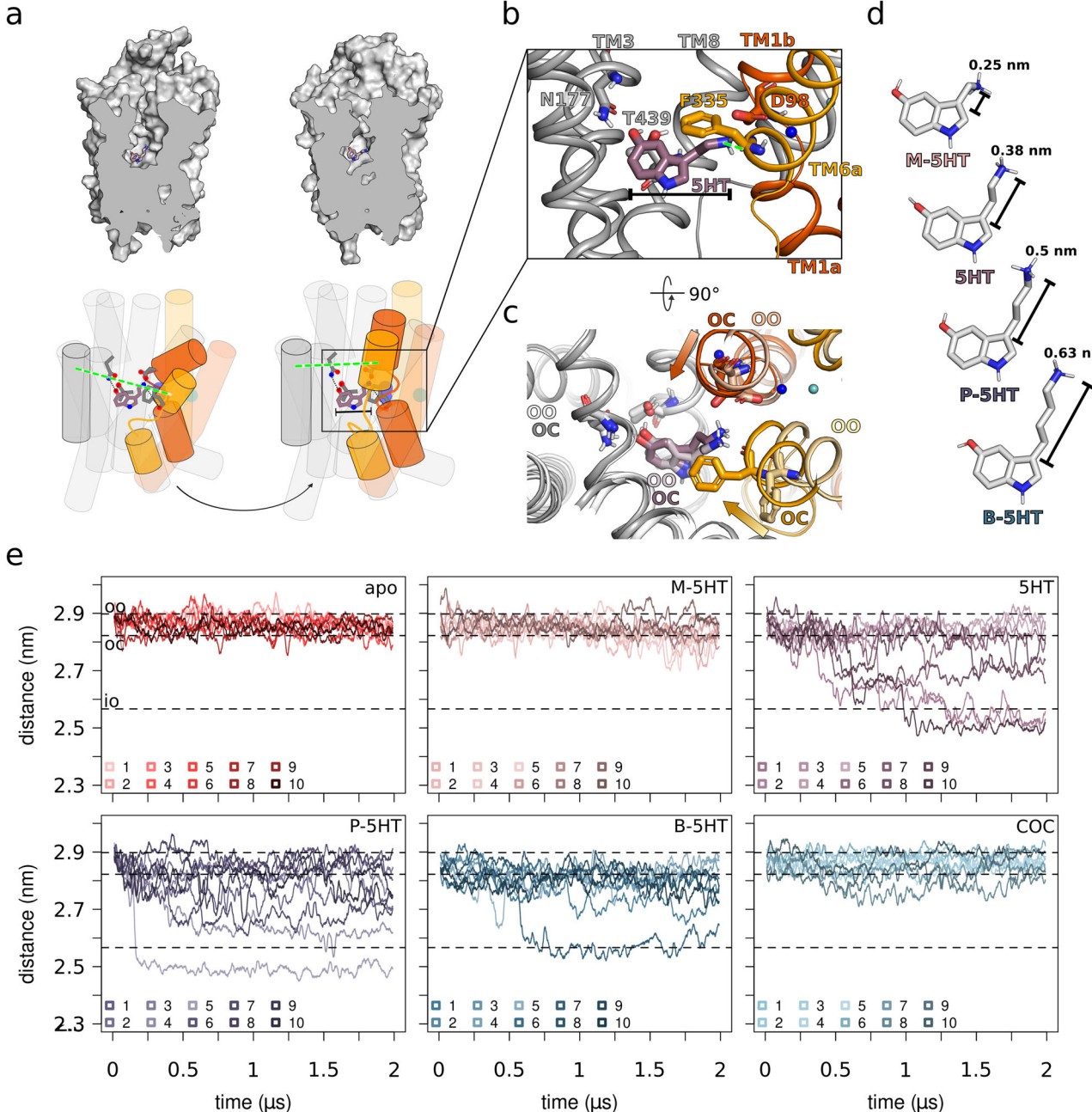

**Fig. 1 | SERT occlusion is triggered by the substrate. a** Substrate-induced occlusion of SERT leads to closure of the outer gate, represented as top) surface and bottom) cartoon. Transmembrane Helix 1 (TM1) and TM2, and TM6 and TM7 are highlighted in red and yellow, respectively. The dashed green line indicates the measured distance across the outer vestibule between TM6a and TM9up. **b** The insert of **a** shows the key interactions between serotonin (5HT) and SERT that trigger the motion of occlusion. **c** Top view of **b** highlighting the displacement of the bundle domain as the transporter moves from outward-open (OO) to outward-occluded (OC), while the scaffold as well as the substrate maintain their positions, coloured opaque and bright, respectively. The red and yellow arrows indicate the direction of TM1b and TM6a associated with SERT occlusion, respectively.

**d** Structure of 5HT and its analogues methyl (M)-5HT, propyl (P)-5HT and butyl (B)-5HT. **e** Time evolution of the distance between TM6a and TM9up as a measure to quantify the degree of occlusion. Each of the six plots displays all ten replicas ($n = 10$) depicted in different colours (apo: shades of red, M-5HT: shades of peach, 5HT: shades of mauve, P-5HT: shades of cyber grape, B-5HT: shades of petrol, and cocaine (COC): shades of light blue). Dashed lines represent the openness of the outer gate by measuring the distances of the centre of mass from TM6a (residue G324 to L337) to centre of mass of TM9up (residue F475 to S477) from 3 known structures: 5I73 (outward open: OO), 6DZV (outward occluded: OC) and 6DZZ (inward open: IO)[18,29,34].

domain. B-5HT preferentially interacts with D98 and F335 at longer distances consistent with the outward-open conformation, thereby shifting the equilibrium further towards the outward-open conformations. Thus, the ability of B-5HT to promote SERT occlusion is further reduced yet remains possible. In contrast, the competitive inhibitor cocaine prevents occlusion by blocking movements of the bundle domain.

To summarise the conformational preferences of SERT in the presence of these compounds and to illustrate the importance of interaction energies between the ligand and F335, we separated the trajectory frames into conformations that show an interaction strength smaller or larger than −20 kJ/mol, respectively (Fig. 2e). 5HT-bound SERT shows a strong conformational preference for occlusion upon establishment of strong interactions. The propensity for occlusion

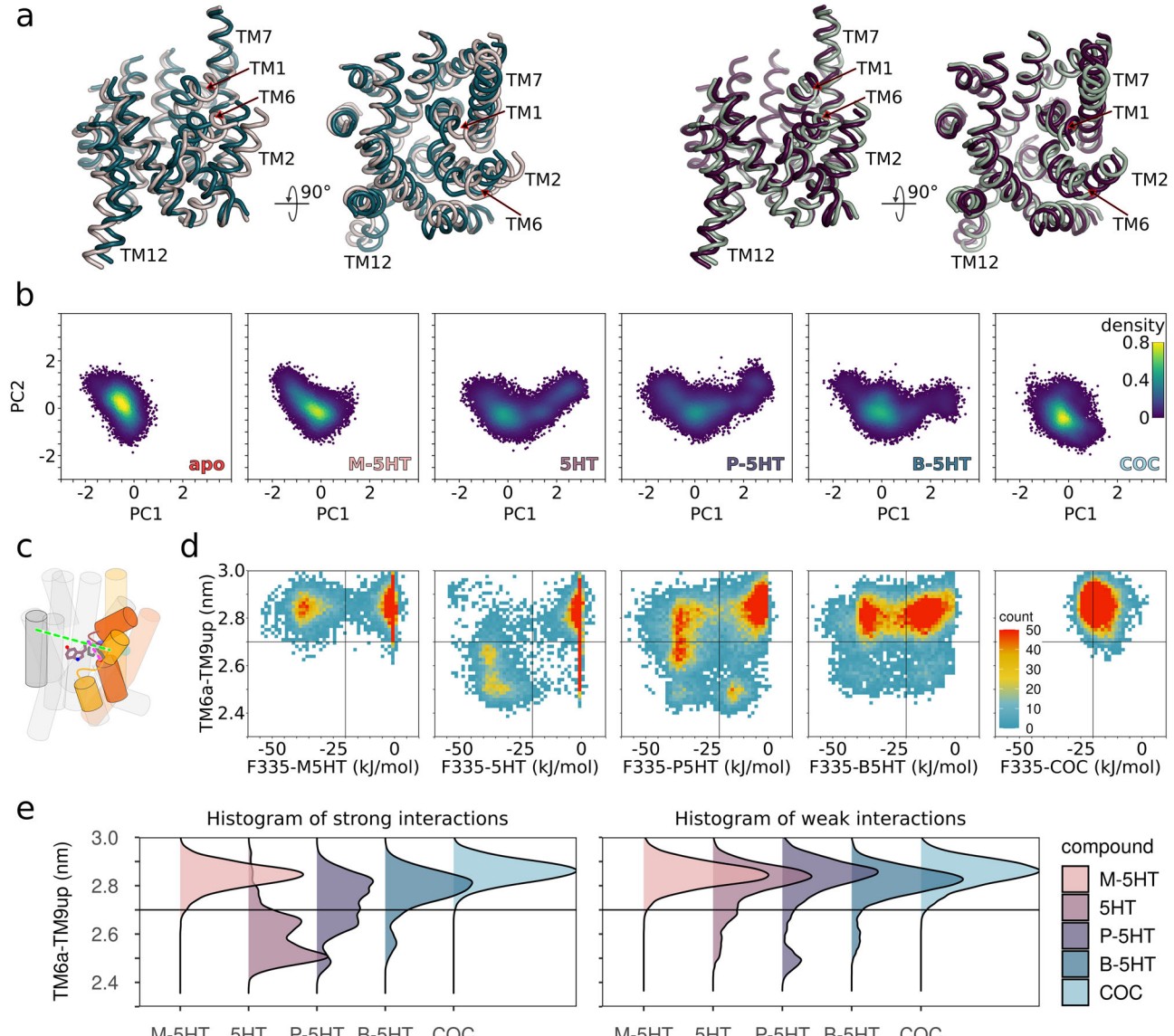

**Fig. 2 | Interactions between compound and the bundle domain support SERT occlusion. a** The two largest principal components (PC) of SERT occlusion of the entire trajectory dataset reveal the dominant motions of occlusion. The images show two extreme conformations of PC1 (left) that are associated with the closure of the outer vestibule and PC2 (right), which represents a rotation of the bundle domain orthogonal to PC1. **b** Projection of each dataset (apo, serotonin (5HT), methyl (M)-5HT, propyl (P)-5HT, butyl (B)-5HT, and cocaine (COC)) comprising 10 trajectories (*n* = 10 each) onto PC1 and PC2. **c** Graphical representation of SERT shown as cartoon: interaction between F335 and compound is depicted as a dashed pink line, and the measured distance between transmembrane helix 6a (TM6a) and

TM9up is shown as a dashed green line. TM1 and TM2 as well as TM6 and TM7 are highlighted in red and yellow, respectively. **d** Correlation of the potential energy of interactions between compound and residue F335 and the degree of occlusion measured as the distance between TM6a and TM9up. The vertical line indicates the separation between the weak interacting (> −20 kJ/mol) and strong interacting (< −20 kJ/mol) conformation. The horizontal line indicates the boundary between the outward-open and the occluded conformation. **e** Direct comparison of SERT conformations as histograms for right) weakly (> −20 kJ/mol) and left) strongly (< −20 kJ/mol) interacting conformations. M-5HT, 5HT, P-5HT, B-5HT, and COC are coloured peach, mauve, cyber grape, petrol, and light blue, respectively.

becomes gradually smaller with the extension of the chain length to P-5HT and B-5HT. In contrast, both the M-5HT-bound and the cocaine-bound SERT remained in the outward-open conformation. These data highlight the importance of the structural geometry as well as the chemical properties of compounds for inducing SERT occlusion.

### Substrate-induced occlusion primes SERT for inner gate opening

Two gates separate the S1 of SERT from either side of the membrane. It is essential for the transport function that at least one of the two gates is closed throughout every step of the transport cycle. Substrate re-uptake by SERT starts with the closure of the outer gate upon substrate binding to the outward-open transporter, which result in the occluded

state. Subsequently, the inner gate opens and SERT releases the co-transported ions and substrate to the cytosol from the inward-open conformation. Here, we investigated substrate-triggered transporter occlusion that leads to the closure of the outer gate. Further, we posed the question, if occlusion would prime SERT for the opening of the inner gate to facilitate substrate translocation. We used Functional Mode Analysis (FMA) to unravel structural changes of the inner gate that are correlated with the interaction between the compound and the bundle domain[39]. FMA transforms the original coordinates into collective modes that maximise the covariance with an input or query function. As substrate-induced SERT occlusion is mediated by inter-actions of 5HT (or analogues) with the bundle domain, we used the respective as a query function. The residue Root Mean Square

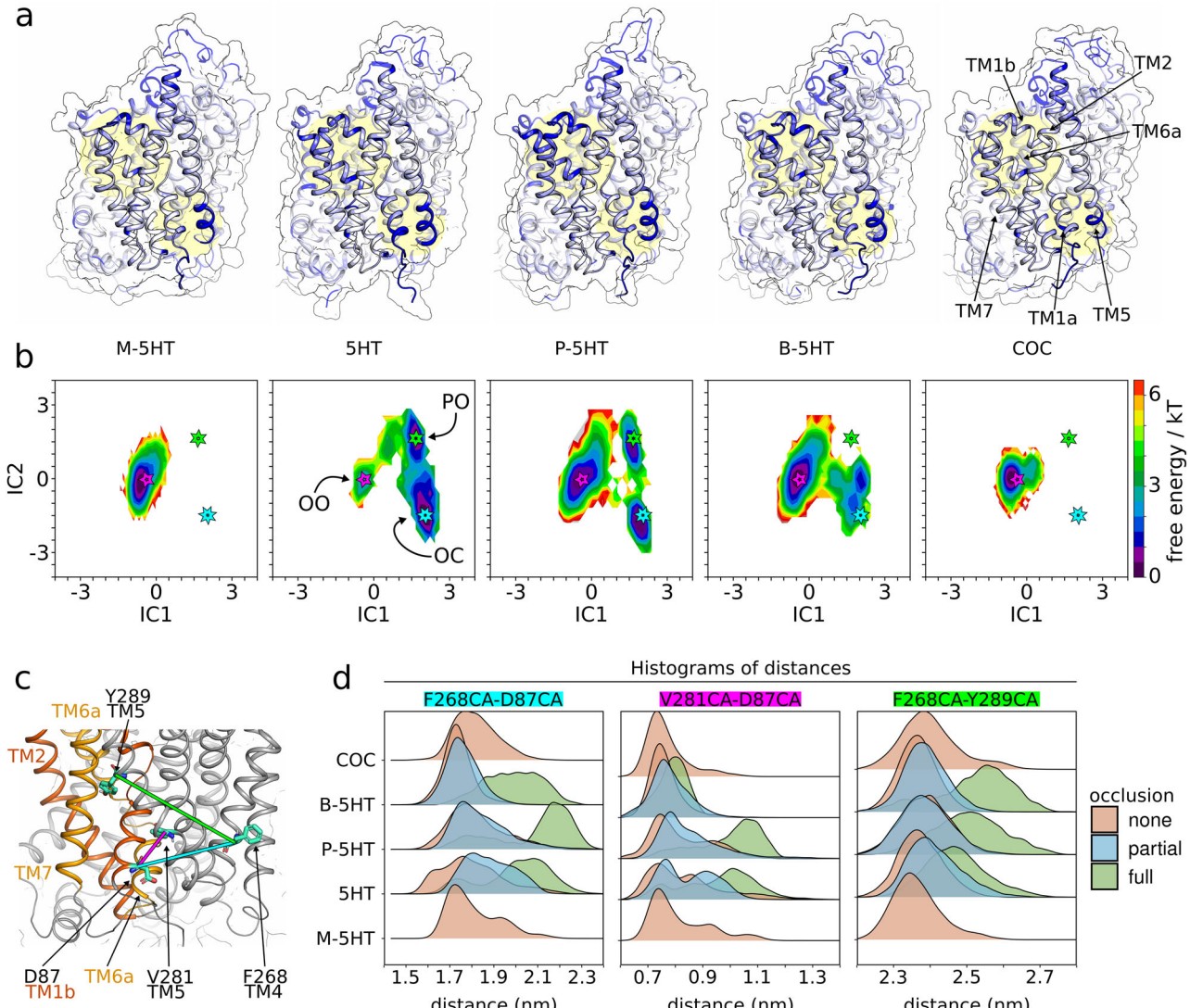

**Fig. 3 | SERT occlusion destabilises the inner gate. a** Mapping of the largest fluctuations (Root Mean Square Fluctuation: RMSF) onto the structure of SERT visualised the regions of largest motions. The darker the blue, the higher the RMSF. Transmembrane helix 1a (TM1a), TM1b, TM6a and TM5a are highlighted by yellow circles in the background. **b** Free energy surface of SERT occlusion using time Independent Component Analysis (tICA). The position of the outward-open, partially occluded and the occluded states are indicated on the plot by a pink, green, and cyan star, respectively. **c** Graphical legend showing the distances measured at the inner gate. TM1 and TM2 as well as TM6 and TM7 are coloured red and yellow, respectively. **d** The frames of the trajectories of 5HT and analogues are separated by the degree of occlusion (none (red), partial (blue), and full (green)). Histograms of distances between D87 (TM1) – F268 (TM4) (cyan); D87 (TM1) – V281 (TM5a) (pink); F268 (TM4) – Y289 (TM5b) (green) quantify the destabilisation of the inner gate.

Fluctuations (RMSF) of the FMA filtered trajectories were projected onto the input structure as β-factors (Fig. 3a). The RMSF corroborated the measures of the degree of occlusion and of the PCA analysis (Fig. 2). Importantly, the FMA could segregate trajectories, which showed a closure of the outer gate from those where SERT remained in an outward-open-like state. As expected, large RMSF values were observed in TM1b and TM6a as the bundle domain tilts during occlusion. Additionally, the first half of TM5 (TM5a) also displayed high β-factors in occluding trajectories, suggesting an allosteric communication (Fig. 3a).

The structural instability of TM5 is a direct consequence of bundle domain motions and therefore of SERT occlusion. TM5 connects the scaffold domain with the bundle domain. While TM5a (residues V274 – F287) moves in concert with the scaffold domain, TM5b (residues P288 – T301) changes conformation together with the bundle domain. Structural instability of TM5 and partial unfolding of the α-helix of TM5a was observed in the inward-facing state of SERT, where this

structural change contributes to the opening of the inner gate[18]. All simulations that reached the occluded state showed an elongation of TM5a as reported by the distance between residue F268 on TM4 and residue Y289 on TM5b (Fig. 3c, d) and a destabilisation of its α-helical structure. In contrast, simulations of the non-occluding cocaine-bound SERT and of the M-5HT-bound SERT remained in the outward-open state and did not show any elongation of TM5a. Similarly, also those simulations of 5HT-bound, P-5HT-bound and B-5HT-bound SERT that did not fully occlude did not show an elongation of TM5a. These structural changes of TM5a are associated with a destabilisation of the inner gate and affect the position of TM1a, as the distance across the inner vestibule increased between TM1a (D87) and TM4 (F268) as well as between TM1a (D87) and V281 (TM5a) (Fig. 3c, d). These measures clearly indicate that occlusion of SERT is sufficient to weaken the inner gate, though we cannot expect to observe complete opening of the inner gate using unbiased simulations, as electrophysiological measurements have shown that it occurs on the millisecond time scale[31].

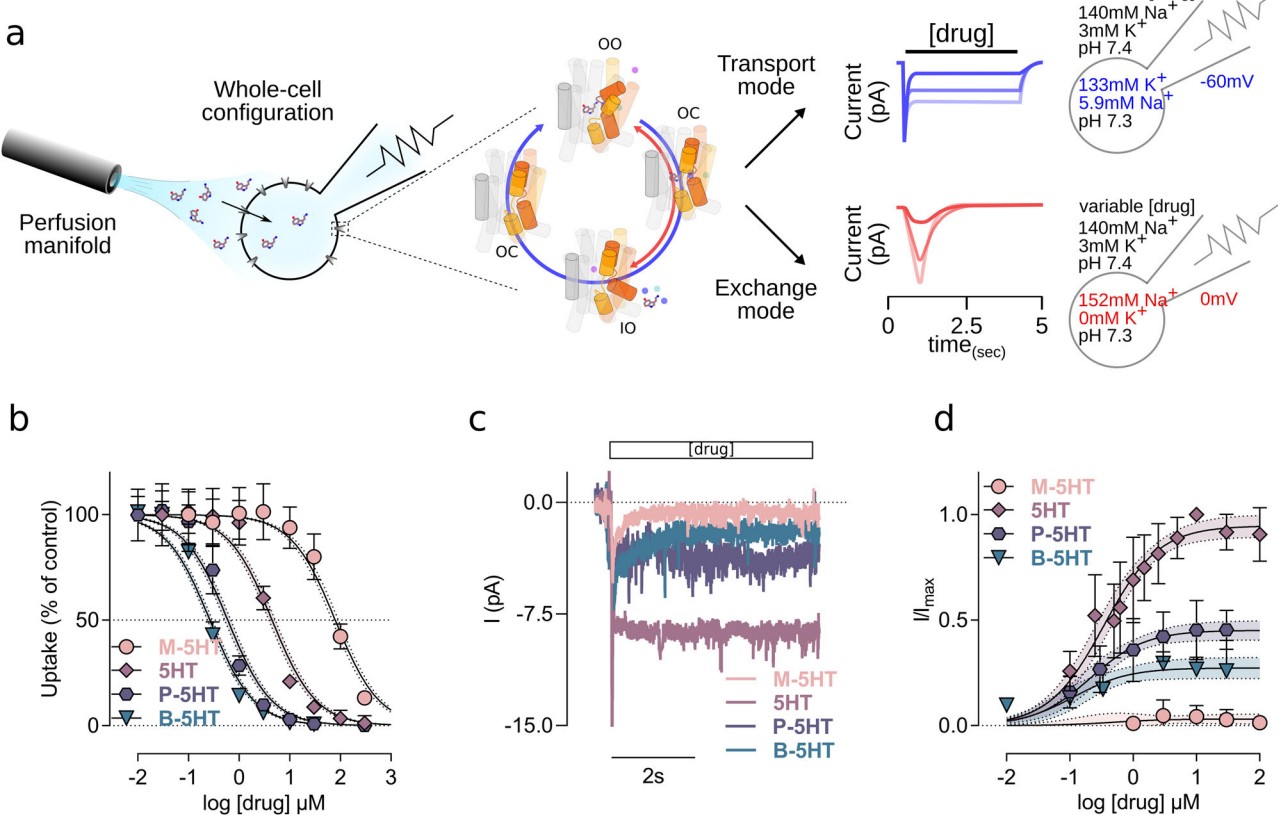

**Fig. 4 | 5HT analogues are transported to very different extents. a** Scheme of the electrophysiological setup, experimental conditions, and the differences between peak and steady state currents. A superfusion system is used for rapid solution exchange of the external solution. Electrophysiological recordings measuring continuous uptake (visible as steady state currents) (blue lines) as a result of continuous cycling of the transporter. The transport cycle with its main conformation (outward-open (OO), occluded (OC), and inward-open (IO) is represented as cartoon. The bundle domain is coloured red (transmembrane helix 1 (TM1), TM2) and yellow (TM6, TM7). The higher the external serotonin (5HT) concentration, the bigger the steady state current (low 5HT: opaque, high 5HT: bright blue). The peak currents (red lines) are visible when uptake is inhibited by inhibitors or by blocking uptake through collapsing of the electrochemical driving forces by identical $Na^+$ concentrations on both sides of the membrane and removing the transmembrane potential by clamping the cell to 0 mV. The higher the external concentration of 5HT, the steeper the peak recovery (low 5HT: opaque, high 5HT: bright). **b** Uptake inhibition showing the effects of methyl (M)-5HT, 5HT, propyl (P)-5HT and butyl (B)-5HT on the SERT mediated uptake of [3H]5HT. Data are as the mean ± SD from $n = 4$ (5HT and B-5HT) and $n = 5$ (M-5HT and P-5HT) independent experiments measured in triplicates. **c** Representative steady state currents elicited by 5HT and derivatives at saturating concentrations (10 μM). **d** Concentration-response relationship of steady state currents induced by 5HT and its derivatives, normalised to a reference 5HT steady-state current elicited at 10 μM. Data are as the mean ± SD from $n = 12$ (M-5HT), $n = 9$ (5HT), $n = 15$ (P-5HT), and $n = 10$ (B-5HT) recordings of independent cells. M-5HT, 5HT, P-5HT, and B-5HT are colour-coded in peach, mauve, cyber grape, and petrol, respectively. Confidence intervals (95 %) of the curve fittings are plotted within ± SD and delimited by dashed lines.

## The position of the alkylamino moiety dictates the free energy surface of SERT occlusion

To quantify the free energy associated with SERT occlusion, we built Markov State Models (MSM) using time Independent Component Analysis (tICA)[40–42]. This method transforms the coordinates into Independent Components (IC), identifies the slowest motions, assigns conformations to microstates to discretise the coordinate space and builds MSMs from the transitions between these microstates, which allowed us to derive a global free energy landscape. We created a common IC coordinate system using the entire dataset and projected each ligand dataset onto the same tICA subspace (IC1 and IC2) to allow us for direct comparison. We assessed the statistical robustness of tICA by quantifying the dependence between the lag time and the implied timescales associated with the slowest processes (Supplementary Fig. 3a) and determined the predictive power of the MSM models using the Chapman-Kolmogorov (CK) test (Supplementary Fig. 3b). We used the bootstrapping technique with replacement to assess the statistical robustness of the free energy landscape by computing the mean ± SD of the energy surfaces (Supplementary Fig. 4).

The most stable state (Fig. 3b) of M-5HT-bound SERT is the outward-open conformation, corroborating the data from the distance measurements, PCA and FMA. Similar data are obtained for cocaine-bound SERT. In contrast, the free energy landscape of 5HT-bound SERT showed that the outward-open state is of high energy relative to the energetically lowest occluded state. The outward-open and the occluded state are of comparable free energy for P-5HT-bound SERT. Although, B-5HT-bound SERT can reach the same occluded conformation as 5HT-bound SERT, the energy balance is inverted, as the outward-open state is of lower energy compared to the occluded state. These results show that the spatial position of the positively charged amine in the S1 has a strong impact on the free energy surface of SERT occlusion.

## The alkyl chain of 5HT has a strong effect on affinity and transport function

To experimentally verify the predictions from MD simulations, we synthesised the 5HT analogues M-5HT, P-5HT and B-5HT (Supplementary Methodology) and pharmacologically evaluated their capability of inhibiting [3H]5HT uptake into HEK293 cells stably expressing the human isoform of SERT (Supplementary Fig. 6). All compounds could completely inhibit [3H]5HT uptake (Fig. 4b), confirming an interaction with SERT, although the $IC_{50}$ values were

dramatically different: while 5HT has an $IC_{50}$ of 4.49 µM [confidence interval: 3.92 µM, 5.12 µM], shortening of the alkyl chain increased the $IC_{50}$ to 79.35 µM [69.53 µM, 90.57 µM]. In contrast, extension of the alkyl chain in P-5HT and B-5HT decreased the $IC_{50}$ to 0.59 µM [0.53 µM, 0.68 µM] and 0.27 µM [0.23 µM, 0.29 µM], respectively. Thus, the potency of 5HT analogues spreads over almost 3 orders of magnitude between the weakest (M-5HT) and strongest interacting compound (B-5HT).

Uptake-inhibition assays provide information on transporter interactions but are inconclusive with respect to their mode of interaction. To differentiate between substrates, partial-substrates, and blockers, we performed whole cell patch clamp electrophysiological measurements. Substrate uptake by SERT moves charges (substrate and co-transported ions) across the membrane, thereby eliciting an initial inwardly-directed capacitive current and a continuous inwardly-directed steady state current (Fig. 4a)[31]. The capacitive current is visible as a sharp peak and is associated with the initial synchronised electrogenic events of binding, occlusion, and the transition to the inward-facing conformation. Both inhibitors and substrates elicit capacitive currents. In contrast, only substrates lead to inwardly-directed steady-state current, which is associated with de-synchronised, but continuous uptake[31].

Example traces (Fig. 4c) of steady state currents of 5HT and analogues applied at saturating concentrations show that 5HT elicits the strongest steady-state current. P-5HT and B-5HT are both partial substrates as they elicit significant lower steady-state (45% and 28%, respectively) currents, while the short M-5HT is almost a pure inhibitor, as it elicits only very small currents (Fig. 4c, d).

## Kinetic parameter for the interaction with SERT

As done previously for the dopamine transporter[43], we have combined radiotracer flux experiments and transporter electrophysiology in living cells to investigate the interaction mode of 5HT analogues at SERT. The electrochemical potential that energises transport can be eliminated by clamping the cell at 0 mV and substituting intracellular $K^+$ with $Na^+$. Representative traces of 5HT elicited isolated peak currents (Fig. 5a) depict capacitive currents. The peak relaxation rate reflects the rate of compound binding, occlusion, and conformational transition to the inward-open conformation as visualised in Fig. 5j, because high internal sodium and 0 mV prevent further steps along the path of the transport cycle. This parameter can therefore be interpreted as flipping rate[44], because it is the slowest step of the process. However, for M-5HT it might be dominated by binding, as SERT does not seem to occlude (Fig. 2b). The flipping rate was smallest for M-5HT 32.06 s⁻¹ [26.34 s⁻¹, 38.86 s⁻¹], highest for 5HT 104.30 s⁻¹ [97.61 s⁻¹, 111.3 s⁻¹] and intermediate for P-5HT 72.30 s⁻¹ [64.47 s⁻¹, 80.70 s⁻¹] and B-5HT 50.05 s⁻¹ [44.21 s⁻¹, 57.09 s⁻¹] (Fig. 5b).

The association rate constants ($K_{on}$) for M-5HT (1.2 µM⁻¹·s⁻¹), 5HT (6.3 µM⁻¹·s⁻¹), P-5HT (26.1 µM⁻¹·s⁻¹) and B-5HT (20.7 µM⁻¹·s⁻¹) are estimated from the y-intercepts of the first derivatives of the non-linear mono-exponential curves fitting of the dose-dependent flipping rate, as done previously[45]. The differences in the $K_{on}$ are much smaller than the differences in $IC_{50}$ of uptake inhibition, suggesting that the modification of the length of the alkyl chain has a smaller impact in inducing conformational changes of SERT (Fig. 5b) than on inhibiting 5HT uptake (Fig. 4b).

The dissociation rate constant ($K_{off}$) from the outward-open conformation can be determined by recording the recovery of the peak current under the same conditions of zero electrochemical gradient. Two consecutive pulses at saturating concentration are used to first bind (and occlude) the test compound, while intracellular release is prevented by high internal sodium. The second time-dependent compound pulse allows for the quantification of the recovery time that is needed to achieve a fully transport competent SERT (Fig. 5j), which under these conditions represent the extracellular release for the

bound substrate. A representative time trace is shown in Fig. 5c. We find that the $K_{off}$ decreases with increasing chain length (Fig. 5d): M-5HT ($K_{off}$=3.16 s⁻¹ [2.01 s⁻¹, 5.23 s⁻¹]) 5HT ($K_{off}$=0.55 s⁻¹ [0.29 s⁻¹, 0.92 s⁻¹]), P-5HT ($K_{off}$=0.31 s⁻¹ [0.22 s⁻¹, 0.41 s⁻¹]) and B-5HT ($K_{off}$=0.15 s⁻¹ [0.15 s⁻¹, 0.24 s⁻¹]), showing a much faster release for M-5HT indicative of a non-occluding SERT.

5HT and its analogues bind to SERT in only one binding mode (Supplementary Fig. 7). To investigate the nature of interaction we performed [3H]5HT uptake assays in the presence of different compounds (Fig. 5e). A hallmark of competitive inhibition is an increase in $K_m$ without effects on $V_{max}$, while non-competitive inhibition reduces $V_{max}$ without changing $K_m$ (Fig. 5f, g). Cocaine is a well characterised competitive inhibitor[46,47], while ibogaine is a non-competitive inhibitor that stabilises the inward-facing conformation[18,19]. We found that pre-incubation with P-5HT and B-5HT at their respective $IC_{50}$ concentration led to non-competitive inhibition while M-5HT showed a mixed competitive behaviour.

## Alkyl chain-length dependent conformational trapping of SERT

Similar to the peak-recovery experiment, steady-state recovery experiments under physiological transport conditions were used to quantify the cumulative $K_{off}$ of substrate from the outward-open as well as the inward-open conformation followed by the return to the outward-open state (Fig. 5j) by applying test compound at saturating conditions (M-5HT at 300 µM, 5HT at 100 µM, P-5HT at 10 µM and B-5HT at 10 µM). A representative time trace for P-5HT is shown in Fig. 5h, the time dependence is shown in Fig. 5i. Cumulative $K_{off}$ values are derived from a mono-exponential fit to these recovered steady-state amplitudes. While the dissociation of M-5HT (0.24 s⁻¹ [0.11 s⁻¹,]) was very fast, P-5HT (0.04 s⁻¹ [0.02 s⁻¹, 0.07 s⁻¹]) and B-5HT (0.03 s⁻¹ [0.02 s⁻¹, 0.04 s⁻¹]) showed similarly slow off-kinetics. Interestingly, the 5HT-mediated steady-state currents after challenging the cells with P-5HT could not recover to the initial steady state amplitude within the 5 min time window. The kinetic parameters are listed in Supplementary Table 2.

## Substrate induced efflux

Drug-induced efflux of 5HT through SERT is typically associated with SERT substrates. We have shown that SERT accumulates in the inward-facing conformation, if challenged with high external substrate[31], indicating the SERT resides sufficiently long in the inward-facing conformation to bind and export 5HT if intracellular concentrations of 5HT and $Na^+$ are sufficiently high. The $Na^+/H^+$ ionophore monensin is used to increase the intracellular $Na^+$ concentration and to dissipate the sodium gradient. Cells, pre-loaded with [3H]5HT and washed by continuous superfusion were challenged with the test compound at its respective $IC_{50}$ concentration in presence or absence of monensin (Fig. 6). M-5HT did not elicit efflux and it could not be augmented by monensin. As expected, 5HT-induced efflux of [3H]5HT was significantly enhanced in the presence of monensin[48,49]. Release of [3H]5HT by P-5HT was similar to 5HT, while no [3H]5HT release above background could be measured for B-5HT. The physiological relevance of the in vitro data was confirmed by measuring the release properties of P-5HT in ex-vivo punches of mouse hippocampal brain tissue preparations, which highly express SERT in its native environment (Supplementary Fig. 8). Efflux could be elicited with P-5HT at 10 µM from the tissue punches loaded with 0.4 µM [3H]5HT. Efflux was SERT specific, as it could be blocked by S-citalopram, a high-affinity and selective serotonin transporter reuptake inhibitor.

## Discussion

Substrate transport by SERT is energised by the transmembrane electrochemical gradient of the co-transported ions $Na^+$, $Cl^-$, and $K^+$[26,27,50]. The conformations and the kinetic scheme of the transport cycle are well characterised[26,28,31,32]. However, i) the interactions that

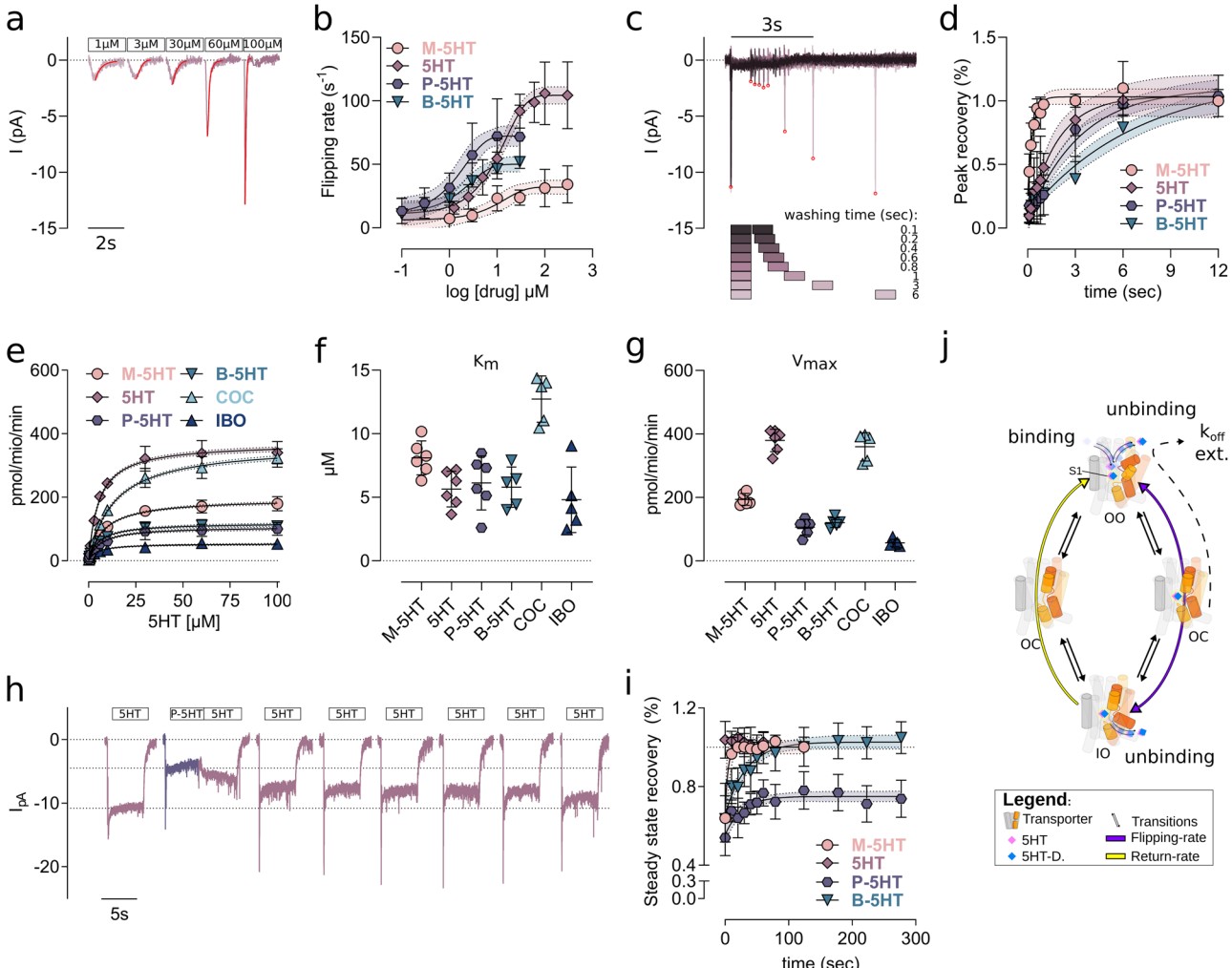

**Fig. 5 | SERT substrate kinetics. a** Representative traces of serotonin (5HT)-induced peak currents at a holding potential of 0 mV. A mono-exponential curve (red line) was fitted to the trace to obtain the relaxation rate. **b** Dose dependent flipping rate deduced from the relaxation rate. Data are represented as mean ± SD from $n = 9$ (methyl (M)-5HT), $n = 20$ (5HT), $n = 16$ (propyl (P)-5HT), and $n = 12$ (butyl (B)-5HT) recordings of independent cells. **c** Representative traces for determining the peak recovery. An initial 5HT peak (100 μM 5HT) was used to define the reference peak amplitude, followed by a second 5HT pulse (of 1 s) with increasing washing interval, indicated by squares. Red circles indicate the maxima of the elicited peaks. **d** Time dependence of the peak recovery is shown as a function of washing time. Data are shown as mean ± SD from $n = 9$ (M-5HT), $n = 7$ (5HT), $n = 11$ (P-5HT), and $n = 8$ (B-5HT) recordings of independent cells. **e** [3H]5HT uptake in the presence of M-5HT (79 μM), 5HT (4.0 μM), P-5HT (0.6 μM), B-5HT (0.27 μM), cocaine (4 μM) and ibogaine (24 μM) at their respective IC50. Data are depicted as mean ± SD of $n = 5$ (B-5HT, COC, and IBO) and $n = 6$ (M-5HT, 5HT, and P-5HT) independent experiments measured in triplicates. **f** Changes in $K_m$ of [3H]5HT uptake upon pre-incubation of the respective compounds represented in **e**. **g** Changes in $V_{max}$ due to the compound pre-incubation, shown in **e**. **h** Reference traces of the steady state

recovery. The initial 5HT (10 μM) pulse (5 s) defines the reference steady state current amplitude. After washout, the test compound (P-5HT) was applied for 5 sec at saturating concentration (30 μM), followed by repeated 5HT (10 μM) pulses and washing intervals. **i** Time course for the recovery of the 5HT steady state current after applying M-5HT (300 μM), 5HT (100 μM), P-5HT (30 μM) and B-5HT (30 μM). Data are shown as mean ± SD from $n = 7$ (M-5HT), $n = 5$ (5HT), $n = 11$ (P-5HT), and $n = 10$ (B-5HT) recordings of independent cells. **j** Graphical representation of the transport cycle, its main conformations (outward-open (OO), occluded (OC), and inward-open (IO)), and transition rates (black arrows). Scaffold is kept in grey, while the bundle domain is coloured red (transmembrane helix 1 (TM1), TM2) and yellow (TM6, TM7). The image indicates the main four steps and highlights the steps probed by experiments: Flipping rate (purple), $K_{off}$ (dashed line) and cumulative unbinding (extracellularly and intracellularly) from the central substrate binding site (S1). 5HT-D include the 5HT-derivatives M-5HT, P-5HT, and B-5HT. M-5HT, 5HT, P-5HT, and B-5HT are colour-coded in peach, mauve, cyber grape, and petrol, respectively. Confidence intervals (95%) of the curve fittings are plotted within ±SD and delimited by dashed lines.

initiate the transport cycle, ii) what triggers SERT occlusion after substrate binding, iii) what primes opening of the inner gate to allow for substrate release to the cytosol, and iv) the physico-chemical and geometrical requirements that define substrates, partial-substrates, inhibitors, and releasers, remain ill-defined at an atomistic scale.

Here we explore the hypothesis that the position of the positively charged nitrogen of the S1-bound substrate (5HT and analogues) is essential to allow for coupling between the substrate and the transporter to initiate the transport cycle. We find essential premises for the transporter SERT to allow for coupling are its ability upon occlusion to

change the distance between the two main substrate interaction sites and to link this change to a destabilisation of the inner gate. Indispensable to a substrate to pull the trigger for occlusion are its ability to strongly bind to the scaffold domain, attract the bundle domain with sufficient strength through electrostatic interactions, to perfectly fit into the S1 only in the occluded state, and to provide the energy for destabilising the inner gate. Inducing SERT occlusion is therefore orchestrated by specific details of several factors in the S1 and centrally involves strong interactions of the positively charged amino group of the substrate with D98 and F335, as well as interactions of D98 with

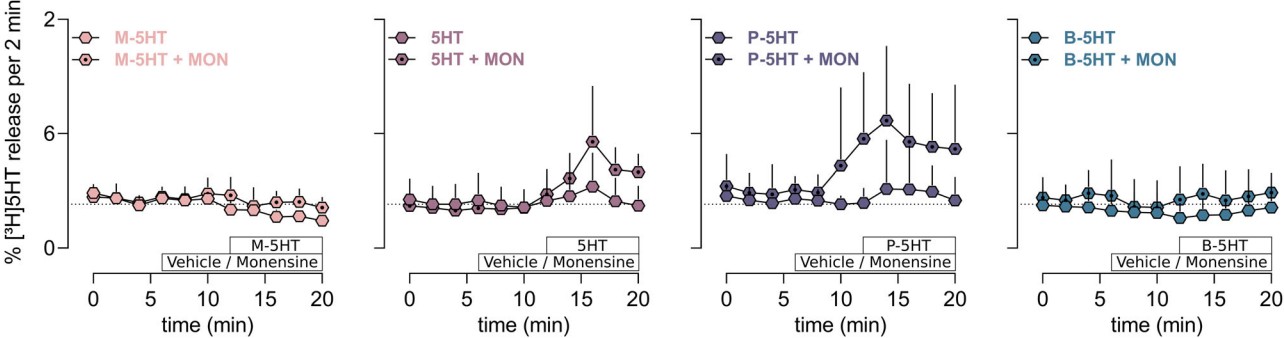

**Fig. 6 | Substrate induces efflux of [3H]5HT.** Traces of SERT-mediated [3H]-serotonin (5HT) release from SERT expressing HEK293 cell upon application of the test compounds at IC$_{50}$ concentration: methyl (M)-5HT at 100 μM, 5HT at 4 μM, propyl (P)-5HT at 0.5 μM and butyl (B)-5HT at 0.25 μM. Data are represented as mean ± SD from $n = 3$ independent experiments measured in triplicates. The first 3 fractions represent basal efflux, then either monensin or the vehicle was added. Afterwards, cells were challenged with substrate alone or in presence of monensin. M-5HT, 5HT, P-5HT, and B-5HT are colour-coded in peach, mauve, cyber grape, and petrol, respectively.

Na1[34]. If the triggering of occlusion was a sole function of the distance between the carboxylate of D98 and the amino group of the substrate, then a D98E mutant of SERT should readily transport M-5HT. The D98E mutant was experimentally investigated in the rat SERT (rSERT)[38] and revealed a more fundamental change in the SERT function, as i) the $K_m$ for Na$^+$ strongly increased and was outside the isotonic range, ii) the reversal potential of the IV curve indicated a switch in ion dependence from sodium in wild type to protons in the D98E mutant, and iii) no current could be detected at physiological pH in case of rSERT D98E. We therefore used MD simulations to investigate the origin of these effects. Our simulation data (Supplementary Notes) show that a repositioning of the carboxylate in the D98E mutant weakens the Na1 binding and disturbs interactions of the substrates' amino group with the F335's carbonyl backbone oxygen atom. We find that these alterations impair substrate induced motions of the bundle domain and therefore interfere with the ligand coupling mechanism that triggers SERT occlusion. Our findings suggest that even though M-5HT, with its shorter alkylamine chain, can compensate for the longer sidechain of D98E by restoring substrate binding affinity through facilitating interactions between TM1 and the substrate, the position of the carboxylate of D98 in the S1 is crucial for the transport mechanism of SERT.

Of the 5HT analogues tested in this study, only the physiological substrate 5HT fulfils all requirements for triggering transport. 5HT binds to the outward-open SERT in a frustrated binding pose. It strongly binds to the scaffold domain (TM3 and TM8), but it is too small for contemporaneously binding to the bundle domain (TM1b and TM6a). Importantly, 5HT maintains spatial proximity between its positively charged nitrogen and the negative charges of the bundle domain (D98 on TM1b and the helical dipole of TM6a) to exert a continuous and sufficiently strong attractive force to pull the bundle domain closer to the scaffold domain to occlude SERT. In contrast, the shorter M-5HT fails to contemporaneously interact with the scaffold domain and the bundle domain in the occluded state. In the outward-open state the gap is too large for establishing a persistent force that would lead to SERT occlusion. Therefore, M-5HT is almost a pure inhibitor. At the other extreme of the congeneric compound series, B-5HT already bridges the distance between the interaction sites at the scaffold and the bundle domains in the outward-open conformation. Binding of B-5HT preferably stabilises an open-like conformation over the occluded state, because of an inversion, as compared to 5HT, of the slope of the free energy surface along the path of occlusion. Like B-5HT, P-5HT also shifted the density of visited conformations towards an outward-open state, as the free energy difference between the outward-open state and the occluded state are small. These changes in conformational preference, caused by shifting the position of the

primary amine by one or two methylene groups, are therefore a consequence of a modified free energy surface along the path of occlusion.

The pharmacological parameters of the congeneric compound series reflect the differences in their interaction with SERT. The mean apparent affinity of M-5HT (79.35 μM), 5HT (4.49 μM), P-5HT (0.59 μM) and B-5HT (0.27 μM), probed by [3H]5HT uptake inhibition assays, showed an increasing potency at SERT with increasing chain length. Similar results were found with C5-fluoro 5HT-derivatives, in which the chain length was increased by up to pentylamine[51]. While the structural changes in substrate geometry had a major impact on the apparent affinity (differed by almost 3 orders of magnitude), it had smaller effects on the $K_{on}$ (~20-fold difference). The cumulative $K_{off}$ (from inward-facing and outward-facing SERT) differed ~8-fold and the flipping rate changed only by ~3-fold. Competitive behaviour of SERT inhibitors, e.g. cocaine, require binding to and stabilisation of the outward-facing conformation[46,47]. Non-competitive ligands, e.g. ibogaine, stabilise the inward-facing conformation, and can therefore not be outcompeted by extracellular 5HT[18,19]. The partial substrates B-5HT and P-5HT are non-competitive SERT inhibitors that cannot be competed by 5HT, suggesting that both compounds dominantly induce structural rearrangements which at least require closure of the outer gate. We infer from the slow return to the outward-open state and the elicited steady state currents that P-5HT and B-5HT unbind slowly to the cytosol. Consistent with the hampered ability of M-5HT to induce SERT occlusion, M-5HT showed a partial competitive behaviour in the inhibition of [3H]5HT uptake, which indicates that 5HT competes with M-5HT from the extracellular side, but also shows that M-5HT retained a limited ability to transition to the inward facing state, which is consistent with the small, yet detectable, steady state current. The in silico data and the in vitro results showed that the window for SERT substrate is very narrow and the change of the alkyl chain by a single methylene affects substrate properties. Similarly, minor modifications of the primary amine as well as the aromatic-moiety result in changes in the substrate/blocker profile[52-54]. From the in silico data we can infer that affinity, $K_{on}$ and $K_{off}$ stem from the ability to interact with both domains, while the flipping rate and substrate transport depends on the ability reaching the occluded state.

The flipping rate is important for substrate re-uptake, as substrate re-uptake can only be achieved if SERT isomerises to the inward-facing state[44]. The four compounds showed similar flipping rates, which indicates that once SERT occludes, the opening of the inner gate only marginally depends on the identity of the ligand. We identified an efficient mechanism of SERT for facilitating substrate translocation, as destabilisation of the inner gate is primed upon occlusion. This may be explained by a mechanism for energy provision by coupling to the

interactions between the substrate and SERT. Before initiating the transport cycle, the outward-facing conformation and the closed inner gate are stabilised by binding of sodium[37]. A ligand with sufficient binding affinity that can also induce transporter occlusion provides the energy to destabilise TM5a and the inner gate. It induces rotation of the extracellular part of the bundle domain that closes the outer gate and the displacement of TM5b, which is the cause of the destabilisation of TM5a through an increase in length, while in parallel the interactions across the inner vestibule weaken to stabilise the closed inner gate.

In short, this study unravelled the physico-chemical basis for discriminating substrates, partial-substrates, and blockers at the molecular level at SERT. Elucidation of the mechanism by which substrates trigger occlusion and thereby initiate the process of translocation to the cytosol opens yet unseen possibilities for rational drug design approaches to selectively target SERT by interfering with one or multiple elements of the substrate-triggered occlusion mechanism.

## Methods

### Model generation
The crystal structure of SERT in the outward-open state with a resolution of 3.15 Å (PDB ID: 5I71[29]) was used as a starting point to generate input systems for molecular dynamics simulations. Further, MODELLER 9.20[55,56] was utilised to generate 100 SERT structures to build the missing side chains and add the co-transported ions[34]. The top 10 structures were selected based on the Discrete Optimised Protein Energy (DOPE) score. Glutamate E508 was neutralised. The insane procedure[57] was applied to embed SERT into a 1-palmitoyl-2-oleoylphosphatidylcholine (POPC) and cholesterol (CHOL) (70:30 mol%) containing membrane.

### Coarse-grained simulations
To equilibrate the membrane around SERT, the MARTINI force field[57,58] was used to simulate the ten systems in coarse-grained (CG) representation. The simulation box ($10 \times 10 \times 10$ nm) was filled with water and 150 mM NaCl. To avoid conformational changes, the protein was restrained and simulated using Gromacs version 2019.2[59] for 1 μs each, ensuring equilibration of the lipid environment.

### Manual docking of 5HT, it's derivatives and cocaine
5HT was docked manually into all-atom outward-open SERT according to our previously described procedure[34] and in accordance with the 5HT-bound occluded structure of SERT[28]. The 5HT analogues M-5HT, P-5HT and B-5HT were oriented in the S1 in the same way with a superimposed indole-moiety. The same method was applied for cocaine, using the crystal structure cocaine-bound Drosophila dopamine transporter (dDAT) as reference (PDB ID: 4XP4[47]). During energy-minimisation and equilibration no major changes in the binding pose occurred, confirming the bound poses.

### Ligand parameterisation
Forcefield parameters for all ligands were obtained for the amber99sb-ildn force field by using the general amber force field (GAFF)[60] and ACPYPE[61]. The charges were derived using the R.E.D. Server[62].

### All atom simulations
After systems equilibrated during the 1 μs GC run, membranes, ions and water were back transformed to an atomistic representation[63]. The membed procedure[64] was used to replace the restrained CG-generated protein with the originally generated transporter conformation to avoid overlapping atoms and to relax the system. Thus, spurious structural distortions introduced by the back mapping procedure are going to be eliminated. The amber99sb-ildn force field was used to describe the protein, water, and ions. The lipids constituting the bilayer were described by Slipid[65,66]. All atom MD simulations were performed by using Gromacs version 2019.3[59]. Each individual system

was energy minimised and equilibrated by conducting a four-step protocol, each step consisting of 2.5 ns long simulation with forces (1000, 100, 10, 1 kJ/mol/nm) acting on heavy atoms, bound ions and the ligand, if present. Initial random and independent velocities were assigned to every system for the MD production runs. For each ligand (5HT-bound, M-5HT-bound, P-5HT-bound, B-5HT-bound, cocaine-bound, and ligand free) 10 MD production runs of 2 μs whereas carried out without position restraints, reaching a total calculation of 120 μs. The Parrinello-Rahmen barostat[67] was used to semi-isotropically maintain 1 bar by applying a coupling constant of 20.1 ps. Temperature (310 K) was maintained by using the v-rescale ($\tau = 0.5$ ps) thermostat[68] by separately coupling the protein with co-transported ions and if present the ligand, the membrane and the solvent. Long range electrostatics were described by the smooth particle mesh Ewald method[69] with a cutoff of 0.9 nm, while the van der Waals interactions were described by the Lennard Jones potentials with a cutoff of 0.9 nm.

### MD data analysis
The MD trajectories were processed and analysed using Gromacs version 2019.3 and MD Analysis version 0.19.2[70,71]. Forces between atoms, residues and ligands were calculated using the Force Distribution Analysis (FDA) package[72]. The Principal Component Analysis (PCA) was performed as previously described[34]. All plots were created using the R package and the python/mathplotlib package. For the PCA the Cα atoms of the transmembrane helices were used as a fitting group. For the Functional Mode Analysis (FMA), all trajectories were fitted to the same reference structure using Cα atoms of the transmembrane helices as a fitting group and all-protein atoms were used for the output. For each trajectory, FMA was performed using 2000 equally spaced frames and using the 10 most correlated components, while as query or input function the potential energy of interaction (kJ/mol) between F335 and the ligand was used. The Markov State Models (MSMs) were constructed using the pyEMMA python package[73]: Firstly, features were defined that describe occlusion and closure of the outer gate (Supplementary Table 1)[34]. Secondly, the conformational space was reduced by performing time-lagged Independent Component Analysis (tICA) with a lag time of 5 ns[40–42]. Thirdly, the trajectories were discretized into 500 cluster centres by applying the k-mean clustering algorithm. Fourthly, the MSMs were estimated using a time lag of 25 ns based on the analysis of convergence of the implied timescales (Supplementary Fig. 3). The free energy surfaces were obtained by using the relation of the stationary distribution (π) value estimated from the transition probability matrix given by:

$$GS_i = -KbT\ln\sum_{j \in S_i} \pi j \qquad (1)$$

Where j denotes the MSM stationary weight of the jth microstate. G, S, Kb, and T represent the Gibbs free energy, entropy, Boltzmann constant, and temperature, respectively. The free energy surfaces were independently derived for every ligand and projected onto the same first and second independent components (IC1 and IC2), which represent the two slowest processes involved in substrate occlusion. The bootstrap method was used to assess statistical robustness by evaluating the mean free energy surface values (Supplementary Fig. 4).

### Chemical synthesis
Unless otherwise noted, chemicals were purchased from commercial suppliers and used without further purification. The purity of the reported compounds is >95% according to NMR. NMR spectra were recorded on a Bruker Avance Ultrashield 400 ($^1$H: 400 MHz, $^{13}$C: 101 MHz) and Bruker Avance IIIHD 600 spectrometer equipped with a Prodigy BBO cryo probe ($^1$H: 600 MHz, $^{13}$C: 151 MHz). Chemical shifts are given in parts per million (ppm) and were calibrated with internal standards of deuterium labelled solvents CDCl$_3$ ($^1$H 7.26 ppm, $^{13}$C 77.16

ppm), DMSO-$d_6$ ($^1$H 2.50 ppm, $^{13}$C 39.52 ppm) and MeOD ($^1$H 3.31 ppm,$^{13}$C 49.00 ppm). NMR assignments of previously not reported compounds were confirmed by $^1$H - $^1$H COSY, $^1$H - $^{13}$C HSQC and $^1$H - $^{13}$C HMBC. TLC was performed using silica gel 60 aluminium plates containing fluorescent indicator from Merck and detected with UV light at 254 nm or stained with potassium permanganate (1 g $KMnO_4$, 6.6 g $K_2CO_3$, 100 mg NaOH, 100 mL $H_2O$ in 1 M NaOH) or anisaldehyde (425 mL EtOH, 16 mL conc. $H_2SO_4$, 10 mL $p$-anisaldehyde, 5 mL AcOH) with heating. HPLC-MS analysis was performed on a Nexera X2 UHPLC system (Shimadzu) composed of LC-30AD pumps, SIL-30AC autosampler, CTO-20AC column oven, and DGU-20A5/3degasser module. Detection was conducted using an SPD-M20A photodiode array and an LCMS-2020 mass spectrometer (ESI/APCI). If not stated otherwise, all separations were performed using a Waters XSelect CSH C18 2,5 μm (3.0 × 50 mm) column XP at 40 °C, and a flow rate of 1.7 mL × min$^{-1}$. Gradient elution was performed using mixtures of HPLC grade MeCN and HPLC grade water with either 0.1 vol% formic acid (acidic separation) or 50 mM $NH_4HCO_2$ solution (basic separation). Flash column chromatography (FC) was carried out with standard manual glass columns using silica gel 60 M (particle size 40–63 μm, 230–400 mesh ASTM, Macherey Nagel, Düren) and solvents as stated in the respective synthetic procedures. GC/MS spectra were measured on a Thermo Trace 1300/ISQ LT (single quadrupole MS (EI)) using a standard capillary column BGB 5 (30 m × 0.25 mm ID). Melting points were determined by a Leica Galen III Kofler and a Büchi Melting Point B−545. An Agilent 6230 LC TOF-MS mass spectrometer equipped with an Agilent Dual AJS ESI-Source was used for HR-MS analysis. The mass spectrometer was connected to a liquid chromatography system of the 1100/1200 series from Agilent Technologies, Palo Alto, CA, USA. The system consisted of a 1200SL binary gradient pump, a degasser, column thermostat, and an HTC PAL autosampler (CTC Analytics AG, Zwingen, Switzerland). A silica-based Phenomenex C-18 Security Guard Cartridge was used as a stationary phase. Data evaluation was performed using Agilent MassHunter Qualitative Analysis B.07.00. Identification was based on peaks obtained from extracted ion chromatograms (extraction width ±20 ppm). Synthetic procedures are provided in the Supplementary Information.

## Materials and chemicals
M-5HT, P-5HT and B-5HT were synthesised by the Mihovilovic lab. All other used compounds, chemicals and supplies were bought from Sigma-Aldrich (St. Louis, MO, USA) and Sarstedt (Nuembrecht, Germany).

## Cell lines and cell culture
HEK293 cells were purchased from ATCC (# CRL-1573; ATCC, USA) and authenticated by STR profiling at the Medical University of Graz (Cell Culture Core Facility), on January 21st 2014. The human wild type (WT) SERT (hSERT WT) was fluorescently tagged by N-terminal fusion with the enhanced yellow fluorescent protein (eYFP)[74]. According to the manufacturer's guidelines the HEK293 cells were transfected by applying the jetPRIME transfection method (VWR International GmbH, Vienna, Austria). 24 h after transfection, geneticin (G418, 200 mg × ml$^{-1}$) was added for 14 days to maintain high selection pressure. After establishing a stable cell line, the G418 concentration was reduced to 50 mg × ml$^{-1}$. First round of cell sorting by using fluorescence-activated cell sorting (FACS, Core Facility Flow Cytometry, Medical University of Vienna) yielded roughly 200,000 cells featuring the desired and consistent expression level of hSERT (Supplementary Figs. 5 and 6)[75,76]. This polyclonal cell line was further cultivated until reaching 90% confluence. Subsequently a second round of sorting was performed by using the same gate setting and yielded roughly 800,000 cells. This stable cell line (YhSERT wt DPS) was used for experiments since showing a highly consistent expression rate in almost 100% of cells (Supplementary Fig. 6). The stable cell line was

maintained in pre-coated 10 cm cell culture dishes in high glucose (4.5 g × L$^{-1}$), L-glutamine (548 mg × L$^{-1}$), 10 % heat-inactivated foetal bovine serum (FBS), 1 % penicillin (100 U × 100 ml$^{-1}$), streptomycin (100 mg × 100 ml$^{-1}$) and G418 (50 mg × ml$^{-1}$) supplemented Dublecco's Modified Eagle Medium (DMEM) and in humidified atmosphere (5 % CO2, 95 % air, 37 °C).

## Flow cytometry
HEK293 cells stably expressing YhSERT WT were dissociated by incubation in 1X Trypsin-EDTA (0.5 g × L$^{-1}$) for 5 min, resuspended in FBS-free DMEM, and the usage of a 40 μm cell strainer (Cornig Nr. 352235). Cells were sorted using a BD FACSAria Fusion Flow Cytometer (BD Biosciences). The BD FACSDiva version 8.0.2 and FlowJo version 10.10.0 software were used to analyse the FACS data. Population 1 (P1) was defined by a forward scatter (FSC)-area (A) and a side scatter (SSC)-A to facilitate the removal of debris. Population 2 (P2) was defined by FSC-A and FSC-height (H) to identify cells of interest characterised by size and granularity, respectively. The final population, population 3 (P3), was based on N-terminally fused eYFP fluorescence intensity, defined by fluorescein isothiocyanate (FITC)-A (488 nm, 530/30 emission filter) and allophycocyanin (APC)-A (640 nm, 670/30 emission filter).

## Radiotracer uptake inhibition assays
The day before the experiment, cells (0.4 × 10$^5$/well) were seeded onto PDL (poly-D-lysine)-coated 96 well plates, while excluding the border wells. At the day of experiment the culture medium was aspirated and replaced with Krebs-HEPES buffer (KHB: 120 mM NaCl, 25 mM HEPES, 3 mM KCl, 1.2 mM $CaCl_2$, 1.2 mM $MgSO_4$, 5 mM D-glucose, pH adjusted to 7.3 with NaOH). After 20 min of equilibration the cells were incubated for 10 min with increasing concentration of the compound of interest diluted in 50 μl. Subsequently, tritiated substrate was added (200 nM [3H]5HT). After 1 min, uptake was terminated by aspiration, cell washing with 200 μl cold KHB and lysation with 200 μl of 1 % sodium dodecyl sulphate (SDS). The lysate was mixed with 100 μl scintillation cocktail (Ultima Gold, Perkin Elmer, Waltham, USA) and radioactivity was determined with a beta-scintillation counter (Perkin Elmer, Waltham, USA). Nonspecific uptake was measured in the presence of 10 μM paroxetine and subtracted. Data was plotted by using GraphPad Prism version 9.4.1 (GraphPad Software Inc., San Diego, USA). IC$_{50}$ values were determined by non-linear regression fits (log(inhibitor) vs. response (Y = Bottom + (Top-Bottom)/(1 + 10^(X-LogIC$_{50}$)).

## Radiotracer combined inhibition and uptake assays
Same experimental preparation as used for uptake inhibition assays. Cells were incubated with concentrations corresponding to their IC$_{50}$ values. After 10 min of preincubation, the [3H]5HT was further added within increasing concentrations to the solution for another 2 min. Subsequent solution aspiration, washing with 200 μl cold KBP and cell lysis with 100 μl Ultima Gold terminated [3H]5HT uptake. Nonspecific uptake was determined in the presence of 10 μM paroxetine and subtracted.

## Preparation of mouse hippocampal punches
Adult male wild type mice (*Mus Musculus*, C57Bl6/N) (12 weeks of age), n = 5, were obtained from Charles River (Sulzfeld, Germany) and sacrificed by cervical dislocation, subsequently the brain was removed and transferred to pre-oxygenated ice-cold Krebs-Henseleit buffer (KHensB: 118 mM NaCl, 4.7 mM KCl, 1.2 mM $MgSO_4$, 1.25 mM $CaCl_2$, 1.2 mM $KHPO_4$, 25 mM $NaHCO_3$, 11 mM D-glucose, pH adjusted to 7.4, supplemented with pargyline 50 μM, nomifensine 0.1 μM and nisoxetine 0.01 μM). The brain was cut into 1 mm thick coronal slices, using Zivic Mouse Brain Slicer Matrices. Hippocampal punches (1 mm in diameter) (Anterior/Posterior: -2.5 to -3.5 mm relative to Bregma) were

collected and stored in ice-cold KHensB until the beginning of the experiment.

Sex-specific effects have not been considered, as the experiment was designed to confirm the concept of efflux in vitro data in native brain tissue.

All animals experiments were conducted in agreement with the ARRIVE guidelines and the UK Animals (Scientific Procedures Act, 1986 and associated guidelines, EU Directive 2010/63/EU for animal experiments) and approved by the national ethical committee on animal care and use (Bundesministerium für Bildung, Wissenschaft und Forschung: BMBWF-2022-0.121.471). Mice were housed in groups of four per cage in a climate-controlled facility, with access to food and water ad libitum. They were kept on a 12-hour light/dark cycle.

### Superfusion release assays with HEK cells
Measurements of time dependent SERT-mediated substrate efflux were conducted as described previously[77,78]. In brief, cells were grown on glass coverslips and preloaded with 0.4 μM [3H]5HT for 30 min at 37 °C. Thereafter, those coverslips were transferred into the superfusion chambers and continuously superfused with KHP (0.7 ml min⁻¹) at 25 °C for 40 min. This is necessary to guarantee and establish a stable basal release. After the collection of 3 2-min-fractions of basal release, cells were either superfused with 10 μM monensin or EtOH (vehicle) for the following 3 2-min fractions. Subsequently, 5 2-minutes fractions were collected by exposing the cells to the compound of interest at $IC_{50}$ concentration. Tritiated substrate collected in 10 ml counting vials was mixed with a scintillation cocktail and quantified by using a beta-scintillation counter. By lysation of the cells with 1% SDS the remaining [3H]5HT was released and collected. Each fraction of released [3H]5HT is represented as a percentage of the total amount of pre-loaded tritiated substrate. 3 independent experiments have been conducted in triplicates.

### Superfusion release assays with brain tissue
Drug-induced [3H]5HT efflux from native tissue was established based on previously published protocols[77,79,80]. Hippocampal punches were exposed to 0.4 μM [3H]5HT for 30 min at 37 °C. One hippocampal punch was transferred into each chamber, followed by a 40-min washout period to establish a stable basal efflux. Samples were collected as 2-min fractions. After the collection of three basal efflux fractions, the punches were superfused with KHensB containing either vehicle (EtOH) or 10 μM S-citalopram for four more fractions, before tissue punches were challenged with 10 μM P-5HT. The experiment was stopped after 24 min. The brain punches were recovered and lysed in 1% SDS and mixed with scintillation cocktail to determine the remaining amount of [3H]5HT at the end of the experiment. [3H]5HT efflux is depicted as fractional rate, i.e. the radioactivity in each fraction is expressed as the percentage of the total radioactivity present at the beginning of each fraction. In the summary plot [3H]5HT-efflux is expressed as the mean of the last four 2-min fractions.

### Whole-cell patch clamp electrophysiological recordings
Procedure to measure 5HT and its derivatives mediated transporter currents have been performed similarly as described previously[44]. In summary, YhSERT WT cells were seeded at very low density in coated PDL 3 cm² dishes a day before the experiment. On the day of experimental examination, the culture medium was replaced by a buffer constituting extracellular conditions ("external buffer": 140 mM NaCl, 20 mM D-glucose, 10 mM HEPES, 3 mM KCl, 2.5 mM CaCl₂, 2 mM MgCl₂, pH adjusted to 7.4 with NaOH). The patch-pipette microcapillary contained a buffer representing different intracellular conditions ("regular internal solution": 133 mM K⁺-gluconate, 10 mM HEPES, 10 mM EGTA, 5.9 mM NaCl, 1 m CaCl₂, 0.7 mM MgCl₂, pH adjusted to 7.2 with KOH; "high-Na⁺ internal solution: 152 mM NaCl, 1 mM m CaCl₂, 0.7 mM MgCl₂, 10 mM EGTA, 10 mM HEPES, pH

adjusted to 7.2 with NaOH). By using an 8 tube ALA perfusion manifold (NPI Electronic GmbH, Germany) and a DAD-12 superfusion system (Adams & List, Westbury, NY) that allows for fast compound perfusion onto the cell and a complete solution exchange around the cell within 100 ms. Compound perfusion was maintained for 1–5 s, depending on the protocol. Passive holding currents were subtracted and the traces filtered by using a 100 Hz digital Gaussian low-pass filter and a 50 Hz harmonics filter. Traces have been analysed using Clampfit 10.2. Data analysis was performed by using GraphPad Prism version 9.4.1.

### Data and statistical analysis
$IC_{50}$, $V_{max}$, $K_m$, $K_{off}$ as well as $K_{on}$ values were plotted and calculated with GraphPad Prism 9.5.1 (GraphPad Software Inc., San Diego, USA). $IC_{50}$ were determined by nonlinear regression, solving equation Y = Bottom + (Top-Bottom)/ (1 + 10^(X-LogEC50)). $K_m$ kinetics were defined by Y = Vmax*X/(Km + X). $K_{off}$ was calculated by Y = Y0 + (Plateau-Y0)*(1-exp(-K*x)) whereas $K_{on}$ was determined by the y-intercept of the first derivative the dose-dependent flipping rate described by: Y = Y0 + (Plateau-Y0)*(1-exp(-K*x)).

### Reporting summary
Further information on research design is available in the Nature Portfolio Reporting Summary linked to this article.

## Data availability
All data, including the source data of the figures as well as each start end structure of SERT, extracted from all trajectories, are available at Zenodo (https://zenodo.org) with the https://doi.org/10.5281/zenodo.10381879. The PDB entry (5I71) for the crystal structure of SERT in the outward-open state with a resolution of 3.15 Å [https://www.rcsb.org/structure/5I71] was used as a starting point to generate MD input structures. The PDB entries for SERT in the outward-open (5I71) [https://www.rcsb.org/structure/5I71], the occluded (6DZV) [https://www.rcsb.org/structure/6DZV], and inward-open (6DZZ) [https://www.rcsb.org/structure/6DZZ] were used to generate reference distances across the outer vestibule. The X-ray structure of Drosophila dopamine transporter in complex with S1-bound cocaine (4XP4) [https://www.rcsb.org/structure/4XP4] was used as a template to manually dock cocaine into outward-open ion-bound SERT. Source data are provided with this paper.

## Code availability
Codes and scripts used to perform analyses can be shared upon request.

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

## Acknowledgements

The research underlying the current publication has been supported by the Austrian Science Fund (FWF) stand-alone grants P32017 and P34670 to T.S., and P35589 to H.H.S., the FWF grants DOC33 (doc.-funds Neuroscience) to H.H.S. and the W1232 (doctoral programme MolTag) to H.H.S. and M.D.M. This project has received funding from the European Union's Horizon 2020 research and innovation programme under the Marie Skłodowska-Curie grant agreement No: 860954 to H.H.S. and T.S. The results presented have been achieved in part using the Vienna Scientific Cluster. R.D.B. was supported by the National Institute of Mental Health, NIH award 5R01MH094527. F.P.M. was supported by a Max Kade Fellowship of the Austrian Academy of Sciences.

## Author contributions

R.G. and T.S. wrote the manuscript with input from all co-authors. R.G., K.S., E.L., M.N., J.M., F.P.M., L.A.S., H.H.S., and T.S. designed experiments, analysed the data, and prepared figures. R.G. conducted all the electrophysiological experiments. R.G. and S.M.C.S. performed radiotracer experiments. R.G. and F.P.M. performed the ex vivo experiments. J.M. performed the in vitro efflux experiments. R.G. built all the systems used for MD simulations and conducted all the MD simulations. R.G., E.L., and L.A.S. performed in-depth analysis of the MD data. E.L. applied PCA and FMA analysis. L.A.S. conducted tICA analysis and generated MSMs. K.S. synthesised all compounds. R.D.B. contributed to the conception and critical revisions. M.D.M., H.H.S., and T.S. supervised the project.

## Competing interests

The authors declare no competing interests.
