## [Peer Review File · Nature Communications]

Ligand coupling mechanism of the human serotonin transporter differentiates substrates from inhibitorsReviewers' Comments:

Reviewer #1:

Remarks to the Author:

In this manuscript the authors reported a compelling computational and experimental study that provides new crucial insights into the molecular mechanism of transport of the serotonin transporter (SERT). By using a set of serotonin (5HT) analogs, characterized by gradual elongation/shortening of the alkyl-linker separating the rigid ring from the amino moiety, the computational analysis demonstrates that 5HT is the one that best stabilize an occluded conformation of the protein upon binding to SERT. Instead, the occluded state becomes progressively less probable by gradual elongation/shortening of the alkyl-linker. Being the occluded state a necessary intermediate of the alternating access transition, these findings elegantly explain how binding of 5HT and coupling ions to SERT optimally promote the protein conformational change and thereby the transport process. Conversely, the other analogs are progressively less optimal substrates, despite some of them have higher binding affinity to SERT compared to 5HT. The computational predictions are conclusively validated by in-vitro and ex-vivo biochemical experiments, which characterize the effect of the alkyl-linker elongation/shortening on the transport function, with measured pharmacological and kinetic parameters. Overall, this work provides an important contribution to our mechanistic understanding of how SERT carries out its function, with main concepts that are arguably transferable to other membrane transporters.

The manuscript is overall very well written and organized. The simulation data is extensive and statistically sound. The analysis thereof is logical, very well done and clearly quantifies the interplay between ligand binding and protein conformational change .

Although the trend is clear, to outline the robustness of the free energy maps in Fig. 3b, I suggest to report an estimate of their statistical error, for example by reporting an average error or an upper bound of the latter based on blocks analysis or any other reasonable error estimator. Other minor corrections/suggestions are: 1) change the term "free energy calculations" with "free energy analysis" on the abstract, as all calculations are based on unbiased molecular dynamics; 2) please double check that ref. 43 is correct on pag. 13 line 335, I didn't notice M-5HT mentioned in that reference; 3) please correct a few typos, e.g. pag. 6 line 121 "was supported the"-> "was supported by the", pag. 19 line 565 "whereas carried out" -> "were carried out", pag. 20 line 566 "restraints ()"; 4) please introduce an appropriate reference that describes nature and structural location of the co-transported ions after mentioning them in the Model generation section.

Reviewer #2:

Remarks to the Author:

In recent years, a number of crystal and cryoEM structures of SLC6 transporters have been described, which has provided considerable detail on the structures of a series of different conformational states that have helped to understand to mechanism of transport. However, one of the major limitations to these studies has been an understanding of how these transporters transition between the different conformational states to facilitate the movement of substrates across the membrane. This study has provided insights into how the serotonin transporter, SERT, transitions between the outward facing conformation to the occluded conformation and then to the inward facing conformation. They have utilised a combination of molecular dynamics simulations, electrophysiological methods and radiolabelled substrate flux measurements to identify the molecular basis of these transitions. There are a couple of key innovations that they have brought to this work. First the use of a set of serotonin-like molecules that allow a controlled look at the key parameters that govern substrate recognition and the conformational changes required for the transition from outward-facing to occluded to inward facing conformations. Second, taking into account multiple changes in the protein structure that allow the transitions to be tracked, and third the combination of these two approaches with correlations of the kinetics of the transport process.

The work is expertly carried out and the conclusions are sound. I do not have any issues with the

experiments presented, but I have a couple of points that I would like to raise for consideration by the authors.

1. I found the terminology "holo" to be a bit confusing. I assume that the term "holo" has been used to describe the transporter in which Na⁺ and Cl⁻ are bound, but not a substrate/inhibitor. At the very least the terminology should be spelled out. If it has no substrate/inhibitor, why have you not used the term "apo"?

2. The use of the set of serotonin-like molecules that differ in the length of the alky chain is a very elegant and powerful approach. Was consideration given to complementing this work with a study of the D93E mutant? It may be predicted that the conservative mutation would shift the functional properties of the transporter such that the shorter M-5HT would be the preferred substrate rather than 5HT, and that 5HT would behave more like P-5HT or B-5HT. Confirmation of the relationship with eh D93E mutant would strengthen the case.

3. The ex-vivo work with hippocampal punches seems a bit out of place and not well integrated into the study. Whilst it is nice to demonstrate that the effects observed can be translated to native SERT, the demonstration of the effect is quite superficial. I don't think that the work is essential for the study because the main body of work is already very convincing, but if the authors feel that it is important to include, then it should be done in more detail. At the very least, you need to compare 5HT with M-5HT, P-5HT and B-5HT, rather than just P-5HT, to demonstrate that the same relationships between the different derivatives hold in native tissue as they do in the recombinant system.

4. There are a couple of minor typographical and grammatical errors that need to be checked.

Point-by-point reply:

We highly appreciated the reviewers' comments, whom we thank for their constructive criticism. We addressed the points, which they raised as follows:

REVIEWER COMMENTS

Reviewer #1 (Remarks to the Author):

In this manuscript the authors reported a compelling computational and experimental study that provides new crucial insights into the molecular mechanism of transport of the serotonin transporter (SERT). By using a set of serotonin (5HT) analogs, characterized by gradual elongation/shortening of the alkyl-linker separating the rigid ring from the amino moiety, the computational analysis demonstrates that 5HT is the one that best stabilize an occluded conformation of the protein upon binding to SERT. Instead, the occluded state becomes progressively less probable by gradual elongation/shortening of the alkyl-linker. Being the occluded state a necessary intermediate of the alternating access transition, these findings elegantly explain how binding of 5HT and coupling ions to SERT optimally promote the protein conformational change and thereby the transport process. Conversely, the other analogs are progressively less optimal substrates, despite some of them have higher binding affinity to SERT compared to 5HT. The computational predictions are conclusively validated by in-vitro and ex-vivo biochemical experiments, which characterize the effect of the alkyl-linker elongation/shortening on the transport function, with measured pharmacological and kinetic parameters. Overall, this work provides an important contribution to our mechanistic understanding of how SERT carries out its function, with main concepts that are arguably transferable to other membrane transporters.

The manuscript is overall very well written and organized. The simulation data is extensive and statistically sound. The analysis thereof is logical, very well done and clearly quantifies the interplay between ligand binding and protein conformational change.

Reply:

We thank the reviewer for the very positive words on our study.

Reviewer:

Although the trend is clear, to outline the robustness of the free energy maps in Fig. 3b, I suggest to report an estimate of their statistical error, for example by reporting an average error or an upper bound of the latter based on blocks analysis or any other reasonable error estimator.

Reply:

There is no direct way to estimate the statistical error of tICA derived Markov State Model. The software package provides several measures to assess the robustness of the model, which we report in the SI Fig. 3: These consist of the relation between lag time and implied timescale (SI Fig. 3a,b) and the Chapman-Kolmogorov (CK) test (SI Fig. 3a,b). We now directly refer to these tests in the results section, not only in the Methods section. Following the suggestion by the reviewer, we implemented a test to estimate the statistical error using the bootstrap methods with replacement and added the results in SI Fig. 4.

To explicitly reference these statistical assessments, we added the following statement: "We assessed the statistical robustness of tICA by quantifying the dependence between the lag time and the implied timescales associated with the slowest processes (SI Fig. 3a) and determined the predictive power of the MSM models using the Chapman-Kolmogorov (CK) test (SI Fig. 3b). We used the bootstrapping technique with replacement to assess the statistical robustness of the free energy landscape by computing the mean \pm SD of the energy surfaces (SI Fig. 4)." This statement can be found in the first

paragraph of the section: “The position of the alkylamino moiety dictates the free energy surface of SERT occlusion”

Other minor corrections/suggestions are: 1) change the term “free energy calculations” with “free energy analysis” on the abstract, as all calculations are based on unbiased molecular dynamics;

Reply:

Please accept our apologies for this incorrectness: we replaced the statement as suggested.

2) please double check that ref. 43 is correct on pag. 13 line 335, I didn't notice M-5HT mentioned in that reference;

Reply:

We thank our reviewer for pointing this out. The reference refers to the method that quantifies the flipping rate. Accordingly, we moved it to the beginning of the sentence.

3) please correct a few typos, e.g. pag. 6 line 121 “was supported the”-> “was supported by the”, pag. 19 line 565 “whereas carried out” -> “were carried out”, pag. 20 line 566 “restraints ()”;

Reply:

We corrected the typos.

In addition, we corrected errors in the English language throughout the manuscript and coloured these grammatical changes in green to distinguish them from the changes in response to the comments by the reviews.

4) please introduce an appropriate reference that describes nature and structural location of the co-transported ions after mentioning them in the Model generation section.

Reply:

Thanks for pointing this out. We added the appropriate reference.

Reviewer #2 (Remarks to the Author):

In recent years, a number of crystal and cryoEM structures of SLC6 transporters have been described, which has provided considerable detail on the structures of a series of different conformational states that have helped to understand to mechanism of transport. However, one of the major limitations to these studies has been an understanding of how these transporters transition between the different conformational states to facilitate the movement of substrates across the membrane. This study has provided insights into how the serotonin transporter, SERT, transitions between the outward facing conformation to the occluded conformation and then to the inward facing conformation. They have utilised a combination of molecular dynamics simulations, electrophysiological methods and radiolabelled substrate flux measurements to identify the molecular basis of these transitions. There are a couple of key innovations that they have brought to this work. First the use of a set of serotonin-

like molecules that allow a controlled look at the key parameters that govern substrate recognition and the conformational changes required for the transition from outward-facing to occluded to inward facing conformations. Second, taking into account multiple changes in the protein structure that allow the transitions to be tracked, and third the combination of these two approaches with correlations of the kinetics of the transport process.

The work is expertly carried out and the conclusions are sound. I do not have any issues with the experiments presented, but I have a couple of points that I would like to raise for consideration by the authors.

Reply:

We thank the reviewer for the very positive words on our study.

1. I found the terminology “holo” to be a bit confusing. I assume that the term “holo” has been used to describe the transporter in which Na⁺ and Cl⁻ are bound, but not a substrate/inhibitor. At the very least the terminology should be spelled out. If it has no substrate/inhibitor, why have you not used the term “apo”?

Reply:

We are grateful to the reviewer for pointing this out to us. We agree with our reviewer and replaced "holo" with "apo", as "apo" is more appropriate for the substrate-free transporter.

2. The use of the set of serotonin-like molecules that differ in the length of the alky chain is a very elegant and powerful approach. Was consideration given to complementing this work with a study of the D93E mutant? It may be predicted that the conservative mutation would shift the functional properties of the transporter such that the shorter M-5HT would be the preferred substrate rather than 5HT, and that 5HT would behave more like P-5HT or B-5HT. Confirmation of the relationship with the D93E mutant would strengthen the case.

Reply:

Following the suggestion by the reviewer, we have carried out extensive simulations (additional 50 μ s) as we repeated the following simulations (apo, M-5HT, 5HT, P-5HT and B-5HT) with the D98E mutant. These data provide essential insights into the effects of the mutation on transport function. The simulation data indicates that extending the acidic sidechain of aspartate 98 by one C-atom (D98E) has various effects on the function of SERT and its conformational equilibrium. These effects extend beyond the intended purpose of shifting the residue 98 COO⁻ and substrate NH₃⁺ pair to achieve a compensatory regain of function for the truncated serotonin derivative M-5HT. We discovered that repositioning of the carboxylate moiety of residue 98 by the D98E mutant i) alters the core of the conformation coupling between the substrate and the bundle domain, ii) facilitates previously impossible interactions between the bundle domain and the scaffold domain, iii) impairs the stability of NaI within the NA1 binding site, iv) interferes with the formation of the outer gate salt bridge, and v) globally redistributes the conformational dynamics of the whole bundle domain. We added an extensive analysis of our simulation data to the supporting information in a new section titled “Simulations of hSERT D98E”. We included a discussion of this new data in the third paragraph of the main manuscript’s discussion.

The D98E rat serotonin transporter (rSERT) mutant was experimentally studied by Barker et al. in 1999 (doi: 10.1523/JNEUROSCI.19-12-04705.1999). These experimental data linked TM1 to substrate binding and transport by revealing an altered SERT function of the D98E mutant. The K_m of Na⁺ increased significantly, exceeding the physiological range. Additionally, the reversal potential was shifted substantially (WT: typical for Na⁺, D98E mutant: typical for H⁺). A relevant observation was that no currents were detected for the D98E mutant under physiological pH conditions, thus

indicating that we would not be able to use electrophysiological approaches to characterise D98E as we used for WT SERT. Consistent with our simulation results, this data indicates a more fundamental change in SERT function. While characterising these changes in transporter function could potentially allow for uncovering new insights into SERT function, we refrain from experimentally characterising hSERT D98E in this manuscript, as we think these experiments are outside the scope of the current manuscript and, in addition, because of experimental challenges due to the absence of detectable currents at physiological pH. Our extensive simulations of WT SERT and the D98E mutant allowed us to overcome these experimental limitations, thus uncovering critical information to understand the substrate coupling mechanism at the molecular level.

Our new simulation data are now discussed in the manuscript as the 3rd paragraph of the discussion.

3. The ex-vivo work with hippocampal punches seems a bit out of place and not well integrated into the study. Whilst it is nice to demonstrate that the effects observed can be translated to native SERT, the demonstration of the effect is quite superficial. I don't think that the work is essential for the study because the main body of work is already very convincing, but if the authors feel that it is important to include, then it should be done in more detail. At the very least, you need to compare 5HT with M-5HT, P-5HT and B-5HT, rather than just P-5HT, to demonstrate that the same relationships between the different derivatives hold in native tissue as they do in the recombinant system.

Reply:

We thank the reviewer for the statements regarding the importance of the ex vivo work. It is relevant to show that our concept also holds true under physiological or near-physiological conditions. However, moving this part of the project to the supplementary material is justified. We do not think that more ex vivo experiments would enhance the overall impact of our study as it would also mean that more animals would lose their lives, which is certainly not in our interest.

4. There are a couple of minor typographical and grammatical errors that need to be checked.

Reply:

We hopefully corrected all the typographical and grammatical errors. We colour the grammatical changes in green to distinguish them from the changes in response to the comments by the reviews.

Reviewers' Comments:

Reviewer #1:

Remarks to the Author:

The authors have thoroughly addressed all my concerns/suggestions and in my opinion the manuscript is now ready for publication.

Reviewer #2:

Remarks to the Author:

I am happy with the modifications made to the manuscript.
Congratulations on a very nice study